


# Are the Rich less Prone to Flooding? A Case Study on Flooding in the Southern Taiwan during Typhoon Morakot and Typhoon Fanapi

Yen-Lien Kuo[1], Ya-Ming Liu[1], Hone-Jay Chu[2], Hung-Ching Lee[1]

[1]Department of Economics, National Cheng Kung University, Tainan, 701, Taiwan
5  [2]Department of Geomatics, National Cheng Kung University, Tainan, 701, Taiwan

*Correspondence to*: Yen-Lien Kuo (yenlien@gmail.com)

**Abstract.** The study uses Taiwan as an example to explore whether the budget allocation of risk reduction depends on income-related political power. Specifically, we empirically examine the effect of household income on the probability of flooding. Beginning in 2006, the government implemented an 8-year project referred to as the "Regulation Project for Flood-Prone Areas" with a budget of NT$115.9 billion (US$3.86 billion). Over half of the budget was allocated to local authorities in southern Taiwan to help them carry out flood risk mitigation projects. As it was not clear how the local authorities set their priorities in allocating their budgets, this study investigates whether high-income individuals may have used their political influence to influence the budget allocation to improve the flood risk reduction facilities in their communities. Villages, whose average household income was within the top 10% in the county or city, were selected as high-income villages and assigned to the treatment group, whereas other villages were included in the control group. The results using propensity score matching (PSM) show that the flood probability of the high-income group (13% and 16.9%, respectively) was lower than that of low-income group (22% and 28%) during Typhoon Morakot and Typhoon Fanapi, suggesting that high-income areas are less prone to flooding, which might stem from their political power.

## 1 Introduction

Global climate change has increased the frequency of extreme weather events, including flooding, which has resulted in huge amounts of resources, in the form of public budgets or private funding, being allocated to reduce the risk and losses from natural hazards. In general, instant downpours of heavy rain caused by typhoons constitute a significant challenge for a hydrological system without a broad area of relatively flat land. Taiwan consists mostly of steep and precipitous terrain and is frequently subjected to intense rainfall. On average in each year, approximately 3.5 typhoons and dozens of torrential rainstorms hit Taiwan, resulting in average annual economic losses of NT$12.8 billion. In recent years, flooding beyond the capital city (Taipei) has become a larger focus of attention of the government, and large budgets have been allocated to reducing the risk of natural hazards in these areas. It is, however, not only unclear how the central government has reached its decisions to distribute its budgets among the various counties and cities, but also how the local authorities, due to the complexity of and specialist knowledge required for flood management, have set their priorities in their budget allocation. This study uses Taiwan



as an example to explore whether the allocation of budgets directed at risk reduction depends on income-related political power. Specifically, we empirically examine the effect of household income on the probability of flooding.

A recent study - Masiero and Santarossa (2021) - had found the occurrence of earthquakes affected the municipal elections. That effect may pass on the government budget allocation in a democratic regime by a certain mechanism. Previous
studies have used "rent-seeking theory" to study corruption and self-interest-seeking behaviors. Krueger (1974) defined "rent-seeking" as seeking to gain economic rent or privileges by manipulating political processes through private resources at the expense of others' interests, e.g., through bribery. Orton and Rowlingson (2007) suggested that wealthy individuals would attempt to act based on their own self-interest by manipulating policy-making processes to shift the focus and benefits of a social welfare policy to themselves. By observing a historic example in Buenos Aires, Argentina in the 1580s, where rural
plots of land were randomly assigned to people, Rossi (2014) found that individuals who were assigned plots of land with higher values (i.e., more wealthy individuals) tended to subsequently gain more political success, thereby demonstrating a significantly positive correlation between income and political power.

Some studies have examined examples of inequality in public resource allocation. Tompkins et al. (2008) found that funds from drought foundations often fell into the hands of high-income households in north-eastern Brazil, that politicians
were inclined to pass short-term but more newsworthy legislation to win more votes, and that, as a result, much needed long-term drought management plans were not implemented in the affected areas. Rasch (2017) suggested that households with high income had, in fact, an adverse effect on the distribution of public resources. They would channel the benefits of public policies for their own self-interest. Adger (1999) observed that sea levees were only made available on coastlines of Vietnam inhabited by high-income households, whereas other inhabitants were not provided with adequate protection against disasters.

Although numerous studies, both theoretical and empirical, have demonstrated the link between political power and self-interest-seeking behaviors, do such findings also apply to the planning of flood risk reduction measures? In other words, would the rich exercise their privilege and political power to gain an advantage in preventing flooding in their communities, eventually leading to the poor facing a different flood probability than that of the rich?

In Taiwan, except for a few major rivers that are managed by the central government, the flood risk management of local
rivers is the responsibility of the respective local authorities. However, due to the low budgets, the flood risk management handled by the local authorities is usually far from satisfactory. Beginning in 2006, the Taiwanese government implemented an 8-year project referred to as the "Regulation Project for Flood-Prone Areas" (the "Project") with a budget of NT$115.9 billion. The Project was provided with a special budget by the central government to help local authorities mitigate floods. Under the Project, the central government was, in principle, responsible for planning and carrying out the construction work
required by the Project. The local authorities could, however, opt to carry out the work themselves.

The budget of the Project was mainly allocated to counties and cities in southern Taiwan. Tainan received the largest share, or an amount totaling almost NT$24 billion, followed by Kaohsiung, Yunlin, Chiayi and Pingtung in that order. More than half of the total budget of the Project was provided to these southern parts of Taiwan. However, in terms of priority and accountability the budget's allocation is still not clear. For example, when a village became flooded, the cause of the flooding





might not necessarily have originated in that village. In a democratic regime, as in Taiwan, the water management authority, whose budget is controlled by the legislature, decides on the final priority of public flood protection, regardless of whether at the central or city/county government level. Since the legislators are elected by votes that are mainly related to the populations of their electoral districts, it needs to be asked whether individuals with higher income use their political influence to gain an advantage in the budget allocation of the Project to reduce the flood risk in their communities when the effects of population

are controlled for.

This study analyzed the data on flooding caused by Typhoon Morakot in 2009 and Typhoon Fanapi in 2010 to gauge whether there has been any effect of the average household income of the villages on the flood risk reduction facilities implemented under the Project since 2006, because the construction of flood risk reduction facilities usually takes at least two years to complete. The factual summaries of the two Typhoons are reported in the Appendix. The remainder of this study is

organized as follows. Sections 2 and 3 describe the data and methodology used in this analysis. Section 4 represents the results, and Section 5 summarizes our findings.

## 2 Data

The maps of the village territories and digital topography were obtained from an open government data platform to determine

the average elevation and the average slope of the villages from the information contained therein. The income data of the villages used in this study were taken from the "2006 Statistical Book of Audited Reported Personal Income for Income Tax" compiled by the Ministry of Finance, Taiwan. The reasons for selecting the data from 3 to 4 years prior to the Typhoons were twofold. Firstly, as flood protection measures usually took a while to complete, any influence of income on the construction of the infrastructure would not have been observed immediately. Secondly, this study investigates whether income would have

had any impact on flood probabilities and not the other way round as in the study conducted by Xiao (2011). In fact, whether the villages were flooded by the Typhoons would not have had any bearing on the average household income of the villages 3 or 4 years before they struck.

The Social, Economic and Geographical Information Services (SEGIS) platform of the Ministry of Interior provides data on population and house prices. Because data on the sold prices of houses and house sales in remote areas were either not

available or there was very little provided on the SEGIS platform either during or before 2012, data on house prices were obtained from sold house prices for the years from 2013 to 2017 and then adjusted by the house price indices to arrive at the weighted average house prices of the villages. Sold house prices published on the SEGIS platform cover the sold prices of 11 types of properties, ranging from traditional apartment buildings (without elevators), shops, commercial buildings, residential buildings, townhouses, high-rise apartment buildings (with elevators), suites, factories, factory offices, farmhouses and

warehouses. Because this study focused on flooding in southern Taiwan, where traditional apartment buildings (without elevators) and townhouses were the two main types of building structures, of which townhouses were more susceptible to flooding, the weighted average sold prices of townhouses were used in this study.





Data on rainfall were obtained from the integrated radar-gauge rainfall estimation, KRID, at the National Science and Technology Center for Disaster Reduction (NCDR). It was the raster data for a 1.25km$^2$ grid. The major flooding caused by
Typhoon Morakot occurred on 8 and 9 August 2009, and that resulting from Typhoon Fanapi on 18 and 19 September 2010. The total rainfall during each Typhoon in the respective two-day period was calculated by using the inverse distance weighted (IDW) interpolation and zonal statistics in the ArcGIS to obtain the average total rainfall for each village.

Data on flooded locations due to Typhoon Morakot and Typhoon Fanapi were obtained from the Disaster Event Records published by the NCDR in Taiwan. Floods caused by Typhoon Fanapi were concentrated in the three southern regions of
Taiwan, namely, Pingtung, Kaohsiung and Tainan. While the floods caused by Typhoon Morakot were also concentrated in southern Taiwan, they were more widespread and also affected parts of central and northern Taiwan, namely, Miaoli, Hsinchu and Keelung. In order to calculate the probability of being flooded in both typhoons, 2,074 valid samples (all villages) in Pingtung, Kaohsiung and Tainan were adopted in this study.

**3. Methodology**

Whether a disaster may strike and its potential impacts are determined by many factors. A traditional risk assessment model was adopted. First, a "hazard" refers to the severity of an impact that a hazard itself may bring and is not attributable to human behavior, e.g., the intensity of rainfall of a typhoon. Second, "exposure" refers to the population or properties that may be exposed to a hazard, e.g., if a typhoon hits a more densely populated area, it will result in more losses. Third, "vulnerability"
refers to all other factors that may affect the eventual impact of a typhoon. There are two categories of vulnerability: the physical/environmental vulnerability, and the socioeconomic vulnerability. The former refers to the physical environments of hazard-affected areas, e.g., the natural terrain in the areas affected, whereas the latter encompasses age, gender, ethnicity, income, and so on. This model had been adopted to access various risk, such as seismic risk in Taiwan (Lin et al., 2015).

For the purposes of this study, whether a village was flooded was the dependent variable. If a village was flooded, the
dependent variable's value would be 1, otherwise 0. The independent variables selected in this study were as follows:

1.   Population: The population exposed in an area is an important variable in a risk assessment model. This study used population data for 2009 and 2010, when Typhoon Morakot and Typhoon Fanapi impacted Taiwan, respectively. We assume that the more populated the area, the higher the priority it would be given on the list of flood risk reduction facilities, because government leaders in a democratic society are concerned with their election campaigns, and hence
these areas should be less prone to flooding.

2.   House price: The house price is an important factor in the probability of flooding (Felsenstein and Lichter, 2014; Hudson et al., 2014). The higher the average house price of a village, the less likely that it will be flooded.

3.   Maximum hourly rainfall: Maximum hourly rainfall is a proxy for the rainfall intensity.

4.   Total rainfall: Total (accumulated) rainfall during a typhoon is one of the factors contributing to a flood hazard. Typhoon
Morakot lasted 6 days in total from 5 to 10 August 2009. The heaviest rainfall and most severe flooding during Typhoon Morakot and Typhoon Fanapi occurred on 8 and 9 August 2009 and 18 and 19 September 2010, respectively. This study





thus uses the total rainfall of the villages for Typhoon Morakot and Typhoon Fanapi during these respective two-day periods.[1]

5.  Elevation: The higher the elevation and the steeper the slope, the less likely that flooding will occur. Thus, this study selected elevation to measure environmental vulnerability.

6.  Income: Income was used to reflect socioeconomic vulnerability. Piketty (2015) suggested that the "rich" could be defined as individuals within the top 10% or 1% in terms of earnings in a community. If the samples had solely been based on those within the top 1% of earnings, this would have resulted in very small samples in this study. Therefore, those individuals within the top 10% in terms of earnings were used in this study. If the average household income of a village was found to be within the top 10% in the relevant county or city, these villages belong to high-income (treatment) group.

---

[1] Maximum hourly rainfall and total rainfall exhibited severe multicollinearity in the case of Typhoon Fanapi. However, this was absent in Typhoon Morakot. To further control the rain elements in this study, both maximum hourly rainfall and total rainfall variables were used in this study.





**Table 1** summarizes the descriptions of the variables used in this study.

| Variables | Detail | Year | | Expected Impact |
|---|---|---|---|---|
| **Typhoon** | | Morakot | Fanapi | |
| **Dependent Variable** | | | | |
| Flooding | Nominal variable: "1" for villages flooded by the Typhoon and "0" for non-flooded villages. | 2009 | 2010 | |
| **Independent Variables** | | | | |
| Population | The population of the villages in the year of the Typhoon. It was assumed that the government would prioritize more densely populated villages over others so far as election campaigns were concerned. | 2009 | 2010 | (−) |
| House Price | The weighted average sold prices of townhouses in the villages from 2013 to 2017 as adjusted by house price indices. | 2013~2017 | | (−) |
| Maximum Hourly Rainfall | The maximum hourly rainfall in the villages recorded during the Typhoon | 2009 | 2010 | (+) |
| Total Rainfall | The total rainfall in the villages recorded during each Typhoon | 2009 | 2010 | (+) |
| Elevation | The average elevation of the villages | 2013 | | (−) |
| Income | Nominal variable: "1" for villages, whose average household income was in the top 10% in the relevant county/city, and "0" for other villages in the same county/city. | 2006 | | (−) |






This study used the logistic regression model and the propensity score matching (PSM) method to estimate the flood probability. PSM had been adopted to control factors affecting flood probability including hazards, exposure and physical vulnerability when comparing household flood damage mitigation measures (Hudson et al., 2014). This study followed the

same theory. The hazard (rain), exposure (population), and vulnerabilities (elevation and house price) were included as confounding variables when analyze the effect of income on flood probability. The empirical model for whether a village flooded during Typhoon Morakot or Fanapi is specified as follows:

$$F_i = \alpha_i + \beta_1 Pop_i + \beta_2 H\_Price_i + \beta_3 T\_Rain_i + \beta_4 Max\_Rain_i + \beta_5 Ele_i + \beta_6 Inc_i + \varepsilon_i$$

(1)

where $i$ represents the village, $Pop$ denotes the population of the village, $H\_Price$ denotes the index of the house price of the village, $T\_Rain$ denotes the total rainfall of the village during typhoons, $Max\_Rain$ denotes the maximum hourly rainfall of the village during typhoons, $Ele$ denotes the elevation of the village, $Inc$ denotes whether the village belongs to the top 10% of villages in terms of income in a city/county, and $\varepsilon$ represents the residual error term.

Propensity scores were calculated through pre-determined confounding variables to find matching units and to assign

units with propensity scores similar to those of the treatment group to the control group (Rosenbaum and Rubin, 1983). Therefore, the effects caused by the confounding variables could be controlled and the treatment effect could be estimated. The first step in the propensity score matching (PMS) method taken in our study was to decide a method for unit matching. The propensity of a village for being flooded was expressed as:

165                $$PF_i = \alpha_i + \beta_1 Pop_i + \beta_2 H\_Price_i + \beta_3 T\_Rain_i + \beta_4 Max\_Rain_i + \beta_5 Ele_i + \vartheta_i \quad (2)$$

The definitions of these variables are the same as those for Equation (1). The residual error term $\varepsilon$ is assumed to have a logistic distribution. $PF$ is an index of the similarity of villages concerning those characteristics in Equation (2). It can be adopted to choose villages with similar characteristics apart from income. The 1:1 nearest neighbor matching method is used in this study,

i.e. the same quantity of observations from each control and treatment group are selected for matching purposes.

The descriptive statistics of the variables for the two typhoons are shown in Table 2. The elevation of the villages is the same for each of these two years. One may be concerned that rich people may move, resulting in lower average income in the flooded area (Smith et al., 2006). We obtained income data for 2006 and 2016 to calculate the income growth rate of the villages. We selected the income of villages flooded by both Typhoons and those not flooded by either from 2,074 available

observations, in order to compare the income growth rates of the flooded and non-flooded villages. The results shown in Table 3 suggest that there is no obvious difference in the income growth rates between the flooded and non-flooded villages. Assuming that the earning ability of any household in the flooded villages is associated with the level of their original income, this result could indicate that the rich did not move away and the poor did not move into the affected villages. In the long term, the income of the affected villages did not decrease after the Typhoons in Taiwan, unlike the research results obtained in a



case study in the USA (Smith et al., 2006). Apart from that study in the USA, it was found that flooding has no long-term

direct impact on population growth either (Husby et al., 2014). As flooding does not seem to be a significant factor affecting

income and the relocation of the residents of the flooded villages in Taiwan, the potential bias caused by the residential sorting

appears to be very limited in this study.

**Table 2** Summary statistics of variables

| Typhoon Morakot  Sample size: 2074 | | | | |
|---|---|---|---|---|
| | Mean | Std. Dev. | Max. | Min. |
| Flood Probability | 0.3351 | 0.4721 | 1 | 0 |
| Village Population | 2647.8 | 2476.6 | 38982 | 120 |
| House Price (NT$ per Ping) | 119562.4 | 55286.2 | 391400.1 | 21963.9 |
| Max. H. Rainfall (mm) | 62.9045 | 13.4580 | 114.4584 | 31.6653 |
| Total Rainfall (mm) | 679.4306 | 166.2727 | 1742.2170 | 311.1189 |
| Elevation (m) | 58.62926 | 157.3282 | 2273.06 | 0.7331 |
| Household Income ($10^3$ NT$) | 691.8264 | 161.5832 | 2089 | 381 |
| Typhoon Fanapi  Sample size: 2074 | | | | |
| | Mean | Std. Dev. | Max. | Min. |
| Flood Probability | 0.2777 | 0.4480 | 1 | 0 |
| Village Population | 2644.2 | 2514.1 | 39640 | 121 |
| House Price (NT$ per *ping*†) | 119562.4 | 55286.2 | 391400.1 | 21963.9 |
| Max. H. Rainfall (mm) | 57.6140 | 19.4350 | 120.4116 | 12.1104 |
| Total Rainfall (mm) | 367.5439 | 166.8045 | 889.4242 | 47.9751 |
| Elevation (m) | 58.62926 | 157.3282 | 2273.06 | 0.7331 |
| Household Income ($10^3$ NT$) | 691.8264 | 161.5832 | 2089 | 381 |

Note: †Areas in Taiwan are often measured in *pings*. One *ping* is equal to 3.30579 m².

**Table 3** Differences in income growth rates between flooded and non-flooded areas

| | Income Growth Rate | | | |
|---|---|---|---|---|
| | **Flooded by both Typhoons (n=261)** | **Not flooded by either Typhoon (n=1063)** | **Difference** | **T Value** |
| Growth Rate | 12.60% | 12.19% | -0.41% | -0.4408 |



## 4. Empirical results

In this section, the first subsection describes the results of the logistic regression as to whether villages were flooded during the two typhoons. The second subsection describes the results of the propensity score matching (PSM).

### (1) Logistic regression

The logistic regression results are shown in Table 4. Total rainfall had a significant positive effect on the flooding probability and elevation had a significant negative effect on the flooding probability in the cases of both Typhoon Morakot and Typhoon

Fanapi. These results were consistent with our expectations. Maximum hourly rainfall had a significant positive effect on the flooding probability in the case of Typhoon Fanapi, but not in the case of Typhoon Morakot. The population of the villages had a significant positive impact on the flooding probability, contrary to our expectations. One possible reason for this could be that the flooding data was collected from self-reported data, and, therefore, the larger the population, the more likely it was that flood incidents were reported. House prices had a significant negative impact on flooding, which was in line with the

findings in the literature (Kousky, 2010; Bin and Polasky, 2004). Income, the key variable in this study, had a significant negative effect on the probability of flooding, which was consistent with our expectations.



**Table 4** Logistic regression results

| Sample size=2074,Top 10% income = 1 | | |
|---|---|---|
| **Variables** | **Morakot** | **Fanapi** |
| Population | 0.000079 *** | 0.0000775 *** |
| | (.0000231) | (0.0000243) |
| House Price | -0.0000169 *** | -0.0000068 *** |
| (NT$ per *ping*) | (0.0000014) | (-.0000012) |
| Max. Hourly Rainfall (mm) | -.0005572 | 0.0163683 * |
| | (.00534) | (0.0063905) |
| Total Rainfall (mm) | 0.0042395 *** | 0.0065651 *** |
| | (0.000749) | (0.0007598) |
| Elevation (m) | -0.0436608 *** | -0.0224764 *** |
| | (0.0038467) | (0.0030573) |
| Income (top 10%) | -0.7015498 *** | -0.7858783 *** |
| | (0.227194) | (0.2220299) |

**Note: Statistical significance (*P<0.1 ,**P<0.05,*** P<0.01)**

**(2) Propensity score matching (PSM)**

Although the distribution of rainfall is truly exogenous, the distribution of high-income villages could still be correlated with elevation and house prices. In order to ensure that the treatment effect of high income was random, the propensity score matching method was adopted to find villages with similar characteristics apart from income. Figure 1 (a) and (b) shows the histograms for the propensity scores after matching. Despite its skewed distribution, there are ample overlaps between the

treated and the control group implying that the matching has successfully retained adequate samples to avoid attrition bias from the cases of off-support. Figure 1 (c) and (d) shows that after matching, the standard percentage of bias across covariates has been reduced to near zero.


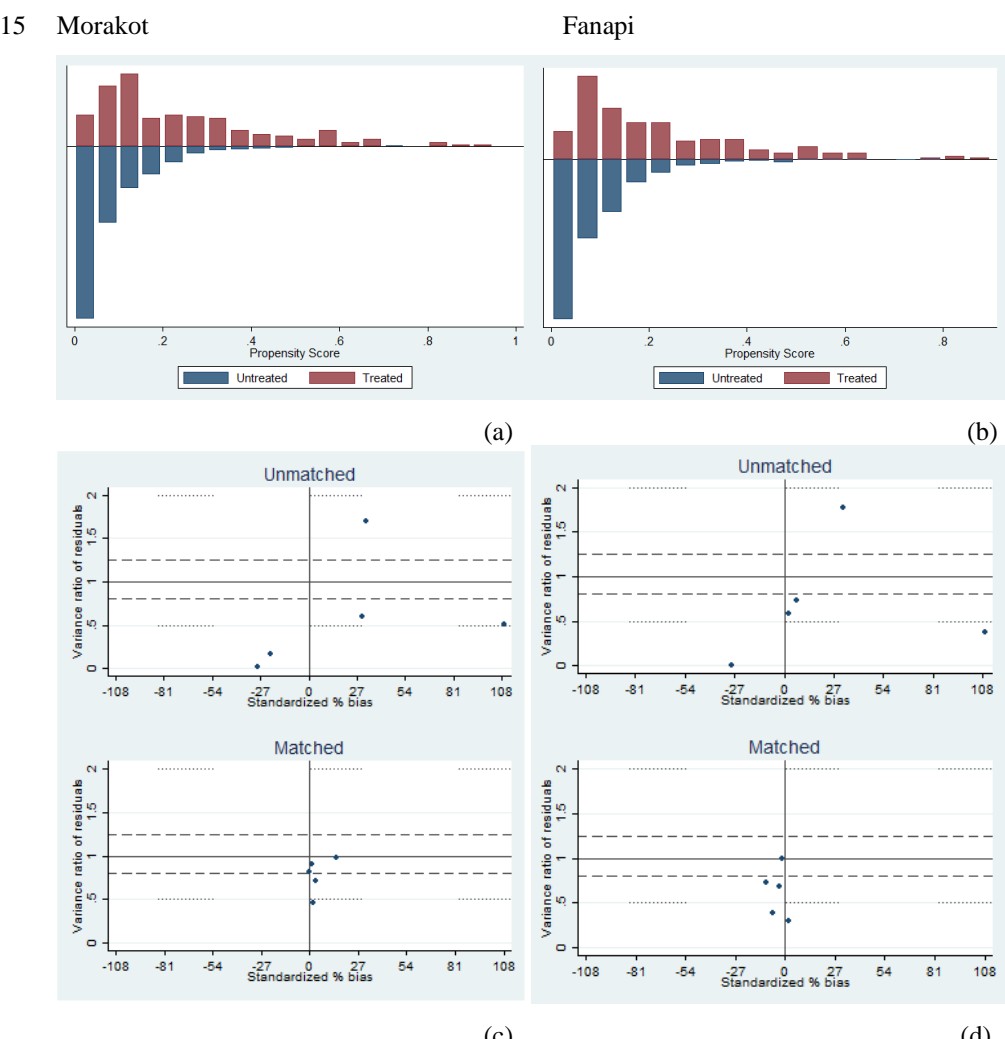

Figure 1 Propensity score matching graphs

Table 5 shows the characteristics of the 207 village samples in each treatment and control group for Typhoon Morakot after matching. The mean of the flood probability of the treatment group was 13.53%, which was much lower than the 24.64% for the control group. The average populations of the treatment and control groups were 3,554 and 3,537, respectively. The average house price of the treatment group was NT$176,196 per *ping*, close to the NT$175,866 for the control group.[2] The average maximum hourly rainfall was 66.1mm for the treatment group, close to the 64.2mm for the control group. The average total rainfall was 653.5mm for the treatment group, and close to 645.5mm for the control group. The average elevation for the

---

[2] One *ping* is equal to 3.30579 m$^2$.





treatment group was 28.2m above sea level, close to the 26.5m for the control group. After using propensity score matching, the characteristics of the villages in the treatment and control groups became very similar. The average household income of

the treatment group was NT$1,025,000 per household, higher than the NT$735,000 for the control group.

**Table 5** Descriptive statistics of the Typhoon Morakot samples (after propensity score matching (PSM))

| | Mean | | S.D. | | Max. | | Mini. | |
|---|---|---|---|---|---|---|---|---|
| | Trt. Group | Control Group | Trt. Group | Control Group | Trt. Group | Control Group | Trt. Group | Control Group |
| Flood Probability | 0.1353 | 0.2464 | 0.3428 | 0.4319 | 1 | 1 | 0 | 0 |
| Village Population | 3554.2 | 3537.1 | 3453.9 | 4093.1 | 29452 | 38982 | 219 | 395 |
| House Price | 176196.2 | 175866.3 | 61831.5 | 60904.7 | 391400.1 | 341835.4 | 52173.1 | 47449.0 |
| Max. H. Rainfall | 66.0997 | 64.2174 | 10.2845 | 10.9371 | 88.3391 | 97.8864 | 45.2654 | 44.8524 |
| Total Rainfall | 653.4965 | 645.4790 | 70.8059 | 89.1945 | 1067.495 | 979.7900 | 444.2594 | 422.6498 |
| Elevation (meters) | 28.2068 | 26.4997 | 14.4969 | 16.1168 | 147.3107 | 132.4810 | 6.5947 | 6.7068 |
| Household Income | 1025.2 | 735.2 | 207.6 | 108.4 | 2089 | 932 | 730 | 438 |

      Table 6 shows the characteristics of the 207 village samples in each treatment and control group for Typhoon Fanapi.

The mean of the flood probability of the treatment group was 16.91%, which was much lower than the 33.82% for the control group. The average populations for the treatment and control groups were 3,426 and 3,493, respectively. The average house price for the treatment group was NT$175,164 per *ping*, almost equal to the NT$174,659 for the control group. The average maximum hourly rainfall was 58mm for the treatment group, almost equal to the 58.9mm for the control group. The average total rainfall was 377mm for the treatment group, slightly lower than the 393.3mm for the control group. The average elevation

for the treatment group was 28m above sea level, slightly higher than the 26.3m for the control group. Therefore, after the propensity score matching, the characteristics of the villages in the treatment and control groups became very similar. The average household income for the treatment group was NT$1,023,000 per household, which was higher than the NT$735,000 for the control group.



**Table 6** Descriptive statistics of the Typhoon Fanapi samples (after the propensity score matching (PSM))

| | Typhoon Fanapi (207 observed values for each treatment and control group) | | | | | | | |
|---|---|---|---|---|---|---|---|---|
| | Mean | | S.D. | | Max. | | Mini. | |
| | Trt. Group | Control Group | Trt. Group | Control Group | Trt. Group | Control Group | Trt. Group | Control Group |
| Flood Probability | 0.1691 | 0.3430 | 0.3757 | 0.0469 | 1 | 1 | 0 | 0 |
| Village Population | 3426.3 | 3493.7 | 3009.1 | 4620.4 | 24356 | 39640 | 202 | 406 |
| House Price | 175164.7 | 174659.0 | 61984.3 | 60494.7 | 391400.1 | 341835.4 | 52173.1 | 51085.7 |
| Max. H. Rainfall | 58.0469 | 58.9436 | 15.2987 | 17.8320 | 111.6155 | 120.4116 | 22.8115 | 19.6113 |
| Total Rainfall | 377.0194 | 393.2966 | 144.7007 | 138.2058 | 727.6195 | 723.0698 | 129.6320 | 97.1310 |
| Elevation (meters) | 28.1308 | 26.3159 | 14.5563 | 28.0940 | 147.3107 | 345.7311 | 6.5947 | 3.3753 |
| Household Income | 1023.1 | 734.6 | 209.2 | 113.5 | 2089 | 1083 | 650 | 478 |

Rubin's B and Rubin's R were also adopted to check the balance of matching. Rubin's B was 21.6 and Rubin's R was 1.08 after applying propensity score matching to the observations for Typhoon Morakot; and Rubin's B was 17.4 and Rubin's R was 1.23 for the observations for Typhoon Fanapi. Rubin (2001) recommended that B be less than 25 and that R be between 250 0.5 and 2 for the observations to be considered to be sufficiently balanced.

As shown in Table 7, the probabilities of flooding for the control groups for Typhoon Morakot and Typhoon Fanapi were 24.6% and 34.3%, respectively, while those for the treatment group were 13.5% and 16.9%, implying that the treatment effect of income reduced the flooding probabilities of Typhoon Morakot and Typhoon Fanapi by 11% and 17%, respectively.






**Table 7** Propensity Score Matching (PSM) results for the top 10% high-income villages

|  | Flood prob. of the Treatment Group (High Income Group) | Flood prob. of the Control Group (Non-High Income Group) | Difference (Treatment Effect) | T -Value |
|---|---|---|---|---|
| Morakot | 0.1353 | 0.2464 | -0.1111 | -2.33 |
| Fanapi | 0.1691 | 0.3430 | -0.1739 | -3.71 |

*Robustness Check*

We also examine whether any selection bias existed in the sampling of the treatment group in this study as a robustness check. Since villages scoring average household incomes in the top 1% as proposed by Piketty (2015) generated too few observations, our sampling of high-income villages was expanded to include villages scoring average household income in the top 20% in the relevant city/county to observe whether PSM results would remain the same for the adjusted "high-income" group. As shown in Table 8, the difference in flooding probabilities between the high-income and non-high-income groups was insignificant in the case of Typhoon Fanapi, but still significant in the case of Typhoon Morakot. The possible explanation

for the difference may stem from the rainfall caused by Typhoon Fanapi being more extreme and concentrated in Kaohsiung city, which caused some flood protection measures to fail.

**Table 8** Propensity Score Matching (PSM) results for the top 20% high-income villages

|  | Flood Prob. of the Treatment Group (High Income Group) | Flood Prob. of the Control Group (Non-High Income Group) | Difference (the Treatment Effect) | T -Value |
|---|---|---|---|---|
| Morakot | 0.1908 | 0.3068 | -0.1160 | -3.89 |
| Fanapi | 0.2729 | 0.2947 | -0.0218 | -0.69 |

**5. Conclusions**

This study has investigated the impact of average household income on the flooding probability of a village in an affected city/county during Typhoon Morakot and Typhoon Fanapi. The results of the logistic regression analysis show that the high-income villages had a lower probability of being flooded during Typhoon Morakot and Typhoon Fanapi. All other control variables including maximum hourly rainfall, total rainfall and elevation were, as expected, significant variables affecting flooding. In order to ensure that the treatment effect of income was random, propensity score matching was adopted to find

villages with similar characteristics. This study found that high-income villages had a lower probability of being flooded. We defined the high-income villages as villages whose average income was in the top 10% of income in the relevant city/county





in southern Taiwan during Typhoon Morakot and Fanapi. Even after changing the definition of high-income villages to those in the top 20% of income in the relevant city/county, the high-income villages still had a lower probability of flooding during Typhoon Morakot.

These results suggest that, between 2006 and 2010, the flood risk reduction may have been concentrated in more wealthy areas, with rich people being likely to attempt to act in self-interest by manipulating policy-making processes and shifting the focus and benefits of a social welfare policy on to themselves (Orton and Rowlingson, 2007), simply because they had more political power (Rossi, 2014). An 8-year NT$115.9 billion budget flood risk reduction project was launched in 2006. We find that, when rainfall and other environmental factors are controlled for, the 2006 high-income villages had a significantly lower

probability of being flooded by the heavy downpours unleashed by Typhoon Morakot and Typhoon Fanapi three to four years after the Project started. High-income people thus might have used their political power to influence the priority of flood risk reduction measures over others to protect their own communities. If, in the future, budget allocations in regard to flood management projects can be made public and transparent, they may deter untoward self-benefiting behavior.

    There were a number of limitations in this study. For instance, the data on the house price variable should have been

taken for the years of the Typhoons or even earlier, just like the data on the population. However, due to the insufficient records of property transactions in remote areas at that time, we used weighted averages of house prices taken from data over a span of many years, which might have differed from actual house sale prices in the years of the Typhoons. The income data covered the period from 2006 and 2016 and were used to investigate household relocation for the treatment and control groups. Even though the average household income of the flooded villages did not reveal a significant difference over this period in our

study, the results might have been different if the income data had covered a longer period. As the data used in this study were collected from the flood data for the two Typhoons, which occurred at the halfway point of the Project, the results of this study may not necessarily have held over a longer period of the Project.

## 6.APPENDIX A

**(1) Typhoon Morakot**

Figure 1 depicts a Typhoon Morakot path map. According to global disaster events published by the National Science & Technology Center for Disaster Reduction (NCDR) in Taiwan, the Central Weather Bureau issued land warnings for Typhoon Morakot at 8am on 6 August 2009. Typhoon Morakot made landfall in Hualian on 7 August 2009 as a moderate typhoon, accompanied by strong southwesternly flows. Typhoon Morakot brought with it unprecedented rainfall in southern and eastern

Taiwan and caused the most davastating flooding in the past 50 years, resulting in severe flooding in Tainan, Kaohsiung and Pingtung, and mudslides on mountainous slopes. A huge landslide destroyed Hsiaolin Village in Jiaxian, Kaohsiung. A total





of 474 people in Hsiaolin Village were buried alive. Typhoon Morakot exited Taiwan close to Taoyuan at 2pm on 5 August 2009. The Central Weather Bureau lifted the typhoon warnings at 5:30am on 10 August 2010.

Figure 2 shows Typhoon Morakot's accumulated rainfall from 5 to 10 August 2010. The heaviest rainfall fell in the mountainous areas of Chiayi, Tainan, Kaohsiung and Pingtung. Most of the rainfall and flooding occurred on 8 and 9 August 2010. The highest accumulated rainfall of 3,060 mm during Typhoon Morakot was recorded in Alishan, surpassing all previous rainfall records in Taiwan and close to the world record. Typhoon Morakot caused an estimated NT$90.47 billion of losses in total. There were a total of 765 square kilometers of flooded areas, including 196 bridges destroyed, 769,159 households without water, 1,595,419 households without electricity, 22,221 households without telecommunications, 516 schools with

building structural damage, NT$19.41 billion in losses to agriculture, forestry, fishery and animal husbandry industries and private facilities, and 1,626 destroyed or damaged buildings and houses.

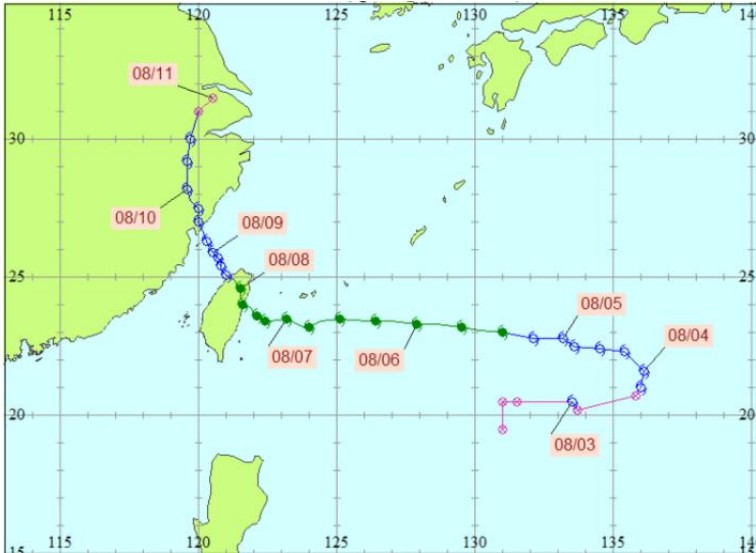

**Figure A1: Typhoon Morakot Path Map**

**Source: The Central Weather Bureau**



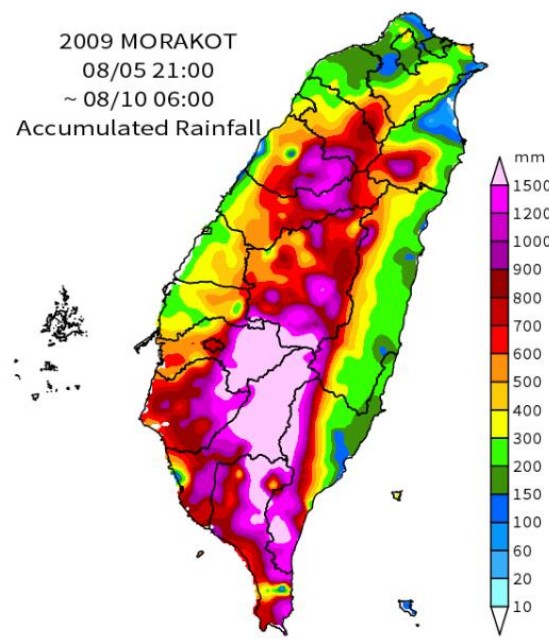

**Figure A 2: Typhoon Morakot Accumulated Rainfall**
**Source: Global Disaster Events (NCDR)**

**(2) Typhoon Fanapi**

Figure 3 depicts the Typhoon Fanapi path map. According to the global disaster events published by the National Science & Technology Center for Disaster Reduction (NCDR) in Taiwan, the Central Weather Bureau issued sea warnings at 11:30 pm on 17 September 2010. Typhoon Fanapi made landfall in Fengbin Township, Hualian County at 8:40am on 18 September 2010 as a moderate typhoon and exited Taiwan from Qigu District, Tainan City at 6:00pm on the same day. Taipower calculated that Typhoon Fanapi caused almost NT$21 billion in losses in total, including power cuts for 905,000 households, two deaths, one serious injury, 110 light injuries, the forced relocation of 6,172 individuals to shelters, 160,000 evacuees, 37 road disruptions, and almost NT$2.1 billion in losses to agriculture, forestry, fishery and animal husbandry industries and private facilities. Even though the damage caused by Typhoon Fanapi was less severe than that wreaked by Typhoon Morakot, as Typhoon Fanapi did not linger in Taiwan as long, Typhoon Fanapi managed to cause flooding in many areas of Tainan, Kaohsiung and Pingtung due to the heavy rainfall, as shown in Figure 4. The main rainfall and flooding caused by Typhoon Fanapi occurred on 19 and 20 September 2010.




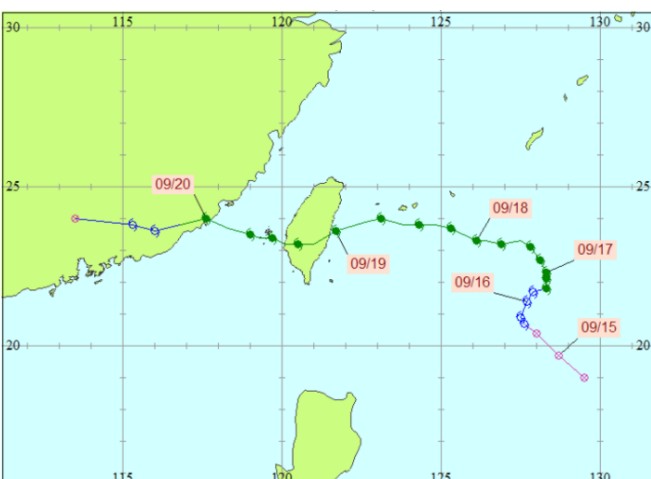

**Figure A3: Typhoon Fanapi Path Map**

**Source: The Central Weather Bureau**

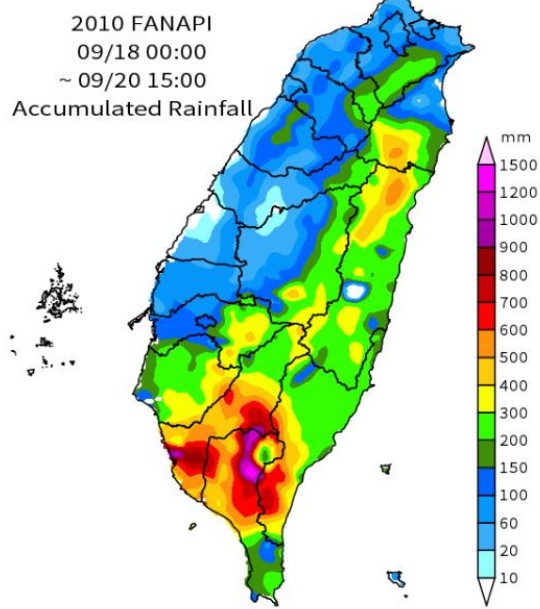

**Figure A4: Typhoon Fanapi Accumulated Rainfall**

**Source: Global Disaster Events (NCDR)**


**Code availability**

The code for this study (logistic regression and propensity score matching) is available on request.

**Data availability**



The data of this study is available on request.

**Author contribution**

Yen-Lien Kuo is the principal investigator of this study. Ya-Ming Liu supervised the Propensity Score Matching analysis. Hone-Jay Chu supervised using geographic information system to sort data adopted in this study. Hung-Ching Lee collected the data and ran logistic regression and propensity score matching analysis.

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
