# Peer review of "Are the Rich less Prone to Flooding? A Case Study on Flooding in the Southern Taiwan during Typhoon Morakot and Typhoon Fanapi"

_Natural Hazards and Earth System Sciences, 2022_

## Author Comment (AC3)

Dear 2 anonymous referees and the editor,

I would like to thank your valuable comments. I had posted my responses right after your comments. In order to make it easier to read, I put my revised responses in the following tables.

Here, I would like to explain in general how the topic of this paper and the major conclusion can be made. According to the literature, there are three kinds of effects between income and flood. Those three effects may make rich people live in low flooding risk areas. The first one which is income affected by flood is not the case. In my response to Referee #1's first comment, words written in italic type, the losses caused by floods are unlikely to affect residents' income even by an extreme typhoon, such as Typhoon Morakot. The second one is by relocation which makes the poor move to higher flood risk areas due to the lower real estate price. Our original manuscript used the income growth rate (see page 7 line 173-180) to explain that this is not the case. Therefore, in order to further avoid another way around effects, whether 2006 income affected flood probability during 2009 and 2010 was tested in this paper.

[Figure]

Three kinds of mechanisms that make rich people live in the low flood risk areas are considered in this study. The first one is that the democratic process sets the priority of flood reduction budget to more populated areas since there will be more votes. The second one was called cost-benefit analysis (CBA) in my response to referees. I may change that if CBA makes confusion. A method called hedonic price method evaluating the benefit of reducing flood risk by calculating the real estate price difference in various flooding probability areas. If this method was adopted, that may divert the flood reduction budget into the areas where high price buildings are located. This method can be further elaborated in the content. Therefore, the population and the house price of the community were adopted as confounding variables in the Propensity Score Matching, and that makes rich and non-rich communities become no significant difference in those two aspects. The third one was called rent-seeking become the most possible mechanism. If the areas where

richer people (10%) reside get priority and reduce the probability of being flooded, the benefit is the reduced expected losses. The most concerning issue of suggesting rent-seeking mechanism is that we don't have the flooding probability before 2006. The reason is that we have to get a large sample size to do this empirical study but the wide spared flooding events seldom happen and the affected regions were not the same except from 2009 Typhoon Morakot and 2010 Typhoon Fanapi. However, whether we have the flooding probability before 2006 may not be an issue as well. The Project was the first project funded by the central government to reduce the flood risk in rivers managed by local governments and before the Project started in 2006 all local governments in Southern Taiwan did not have enough flood reduction budget (see page 1 line 54-60). That is another reason why this study should be published. After this eight-years project started in 2006, a series of huge budget flood reduction plans kept conducted but the budget allocation is still mysterious. In order to avoid misunderstanding that this result had been proven as a long-term phenomenon, the topic of this paper can be changed to 'Are the Rich less Prone to Flooding during Typhoon Morakot and Typhoon Fanapi in the Southern Taiwan?'

Other issues had been proposed by reviewers including what type of construction in the project, the definition of flooding, the luxury dwellings during inundations in Taiwan, introducing hedonic price method on flood reduction benefit assessment, the motivation of rich people to reduce flood loss will be added, further explain or revise in the main content.

Sincerely yours,
The corresponding author

| Anonymous Referee #1 | Authors |
|---|---|
| The authors state that "high-income individuals may have used their political influence to influence the budget allocation to improve the flood risk reduction facilities in their communities" (Abstract, and Page 15, Lines 286-287). That is quite a statement, that requires strong evidence. The statement would require that 1) flood risk has actually decreased in those high-income areas, and 2) that the flood risk also has been reduced *more* | We did not have the flooding probability of villages before the project. However, as the title of this study, we did proof that those 2006 high income (10%) villages had less flooding probability than 2006 non-high income villages during 2009 and 2010 typhoons in Southern Taiwan. Rent-seeking is one of the reasonable and possible mechanism because the village's rainfall is totally exogenous and the rainfall, terrain, population, and house price of village |

in areas with higher incomes, compared to areas with lower incomes. However, neither of these is shown in the analysis.

The only thing the authors show is that there is a difference between income and flood risk. But this is well-known from past research in developed countries as well as developing countries. Lower income households settle in locations that are more flood prone, for several reasons, often a higher flood risk also leads to lower property prices, leading to poorer populations to move here.

I do not doubt that mechanisms of political influence, and nontransparent processes are at play in Taiwan. However, the current study simply cannot deny or confirm any of that to have an effect on actual reduction of flood risk.

Answering the central claim from the paper would require an analysis of the flood hazard before and after the programme, to analyse whether there is any *difference* in flood risk reduction for the different income groups. So how was the flood probability of the communities before the programme that started in 2008? The authors cannot show that.

In Tables 5 and 6, in fact some of the effects of the location choices that I refer to can be seen. In particular, elevation plays a role here (and is related to flood probability, as seen in

were paired by PSM to be no significant difference between high income and non-high income villages. We had used T-test to check the mean difference of variables of treatment group and control group was insignificant including elevation. The T-test results can be added to be an appendix. Rubin's B and Rubin's R were also adopted to check the balance of matching and fitted with its standard. Since the risk reduction efforts toward more population and high real estate price area are democratic and economic (cost-benefit analysis) mechanisms, respectively, rent-seeking is possible mechanism.

Concerning flooding causing migration (poorer population move to higher flood risk area), the difference of income growth rates between 2006 to 2016 of flood prone villages (flooded both during 2009 typhoon Morakot and 2010 typhoon Fanapi) and non flood prone villages were insignificant. Please check Page7, Lines 173-178. As flooding does not seem to be a significant factor affecting income and the relocation of the residents of the flooded villages in Taiwan.

Concerning flooding reducing income, typhoons in 2009 and 2010 **cannot** deteriorate 2006 income. Besides, the following losses estimation and the vicim's survey of Typhoon Morakot showed the damages suffered by victim households were not huge.

Table 4), with the low-income group having a lower elevation, and thus potentially a higher flood hazard.

Also, I wonder about the uncertainty of the flood probability estimates. The authors report that this is collected from self-reports (Line 198), but how could this affect the analysis?

Additionally, the authors cannot exclude the possibility that floods from typhoons had effects on income, as they suggest also themselves on page 3 (Lines 84-87). Although the income data is from 2006, the authors also report that several typhoons hit Taiwan every year, and such impacts could affect incomes, so this could in fact be an additional factor, as shown also in other studies (e.g. https://doi.org/10.1016/j.ecolecon.2020.106879 and https://doi.org/10.1016/j.jenvman.2022.114852).

Finally, I have reservations about whether the programme has led to such investments that there would be a noticeable effect on flood risk for these two specific events. $3.86 billion seems a lot, but it also seems this was spent on quite a large area, and both events were quite extreme.

Moreover, the limited description seems to imply that most of the implemented measures would actually benefit several riparian communities, such as "construction works" that suggests

*"There were 140,424 households with flooding depths of more than 50 cm during Typhoon Morakot according to an investigation report conducted by the Typhoon Morakot Post-Disaster Reconstruction Commission of the Executive Yuan, Taiwan. A total of NT$5.31 billion in damages nationwide and an average of NT$37,814 per household were caused by Typhoon Morakot according to the 2009 annual report of the NCDR. Comparing those to the average annual household income of NT$1,074,180 in 2009, the damages suffered by victim households were not huge. Lastly, changes in income after the disaster were investigated. According to the "Social Impacts and Recovery Survey of Typhoon Morakot (Phase 1)" conducted by the NCDR, where a questionnaire survey was carried out on Typhoon Morakot victims (i.e. households whose houses were so severely damaged that they had become uninhabitable), income of 56% of the victims remained unchanged, whereas 17.9% of the victims showed income increases and 25.4% income decreases. The unemployment rate of the affected households increased by 4.2%. Overall, flooding did not cause too severe an impact on household income."*

Those two events were quite extreme. Typhoon Morakot is the most serious typhoon (the highest losses) in the

| | |
|---|---|
| structural flood protection, such as levees and reservoirs. Or are there any engineering reasons why the measures would have benefitted certain geographic locations, and not others? The current description is highly suggestive (Lines 54-70), but lacks factual descriptions of what investments and construction works were made.

In sum, I think the main conclusion from the paper is not supported by the research design and the results. The authors only show that the lower income communities have a higher flood risk. | history of Taiwan. The losses caused by other smaller events during 2006 to 2010 were much smaller than that by typhoon Morakot.

More than half of the total budget of the Project was provided to these southern parts of Taiwan. The budget was mainly for structural flood protection, such as levees, pumping stations, and detention ponds. Almost all rivers already had some sort of levees before the project. Due to the Project, the local governments decided the priority and the allocation of enhancing levees and building detention ponds. We used a community/village which is the lowest administrative entity to have a large sample size.

At least, studies of social vulnerability to flooding concerned the poor but this study analyzed 10% high income villages. PSM had been adopted for the first time to find villages with similar rainfall, population, house price, and terrain, and found that high income villages are less prone to flooding during 2009 and 2010 typhoons. |

| | |
|---|---|
| First, the auhtors state that "Rent-seeking is one of the reasonable and possible mechanism because the village's rainfall is totally exogenous and the rainfall, terrain, population, and house price of the village were paired by PSM to be no significant difference between high income and non-high | Indeed, rent seeking is defined as that the act of obtaining special treatment by the government to create economic profit. Economic analysis is based on the status quo. The Project is the first project funded by the central government (see page 1 line 54-60). If the areas where richer people (10%) |

| | |
|---|---|
| income villages." First of all, I think that rent-seeking is a term used normally for more direct benefits from e.g. subsidies, or other special treatment by the government. I am not sure if benefits of risk reduction investments really includes this. But I am not an economist. | reside get priority and reduce the probability of being flooded, the benefit is the reduced losses. |
| Second, the term "migration" is not mentioned by me. I am not sure why the authors bring this up. | You mentioned that the poorer population move to higher flood risk area due to the lower real estate price. Our original manuscript used the income growth rate (see page 7 line 173-180) to explain that this is not the case. High income people live in low flooding probability areas because of migration. |
| Third, the authors write that: "more population and high real estate price area are democratic and economic (cost-benefit analysis) mechanisms, respectively, rent-seeking is a possible mechanism."

I have two issues with this statement. If the process was democratic, then rent seeking would not be a problem. This seems to contradict the main statement from the authors, that the process is in fact not doing justice to welfare, or equity, and is therefore not democratic.

Also, real estate prices are not used for cost-benefit analyses to decide on measures to reduce risks from natural hazard. It is damage costs, or more precisely, replacement and repair costs in the event of a flood. This is highly | The statement may not be clear enough. The democratic mechanism will send the budget to more populated areas because of more votes. It is one of ways to estimate the benefit of flood reduction. Using real estate price is the one called the hedonic price method. The key term of that is marginal implicit rent which is the rent differences between various flooding probability in a real housing market. This method was first proposed by MacDonald et al. (1987) and keeps being adopted Yang (2008) in Taiwan and nowadays around the world, i.g. Egbenta et al. (2015). We can add the description of hedonic price method and why that is regarded as cost-benefit analysis of flood reduction into the manuscript. |

| | |
|---|---|
| problematic, as I also write in my comment on the author response to RC2, below. | |
| Fourth, and finally, the authors write that "Concerning flooding reducing income, typhoons in 2009 and 2010 can deteriorate 2006 income". But then later: "Nevertheless, 2009 and 2010 typhoons cannot affect 2006 income." I am not sure if I can follow that. | It was a typo. I revised that. |

---

## Author Comment (AC4)

Dear 2 anonymous referees and the editor,

I would like to thank your valuable comments. I had posted my responses right after your comments. In order to make it easier to read, I put my revised responses in the following tables.

Here, I would like to explain in general how the topic of this paper and the major conclusion can be made. According to the literature, there are three kinds of effects between income and flood. Those three effects may make rich people live in low flooding risk areas. The first one which is income affected by flood is not the case. In my response to Referee #1's first comment, words written in italic type, the losses caused by floods are unlikely to affect residents' income even by an extreme typhoon, such as Typhoon Morakot. The second one is by relocation which makes the poor move to higher flood risk areas due to the lower real estate price. Our original manuscript used the income growth rate (see page 7 line 173-180) to explain that this is not the case. Therefore, in order to further avoid another way around effects, whether 2006 income affected flood probability during 2009 and 2010 was tested in this paper.

[Figure]

Three kinds of mechanisms that make rich people live in the low flood risk areas are considered in this study. The first one is that the democratic process sets the priority of flood reduction budget to more populated areas since there will be more votes. The second one was called cost-benefit analysis (CBA) in my response to referees. I may change that if CBA makes confusion. A method called hedonic price method evaluating the benefit of reducing flood risk by calculating the real estate price difference in various flooding probability areas. If this method was adopted, that may divert the flood reduction budget into the areas where high price buildings are located. This method can be further elaborated in the content. Therefore, the population and the house price of the community were adopted as confounding variables in the Propensity Score Matching, and that makes rich and non-rich communities become no significant difference in those two aspects. The third one was called rent-seeking become the most possible mechanism. If the areas where

richer people (10%) reside get priority and reduce the probability of being flooded, the benefit is the reduced expected losses. The most concerning issue of suggesting rent-seeking mechanism is that we don't have the flooding probability before 2006. The reason is that we have to get a large sample size to do this empirical study but the wide spared flooding events seldom happen and the affected regions were not the same except from 2009 Typhoon Morakot and 2010 Typhoon Fanapi. However, whether we have the flooding probability before 2006 may not be an issue as well. The Project was the first project funded by the central government to reduce the flood risk in rivers managed by local governments and before the Project started in 2006 all local governments in Southern Taiwan did not have enough flood reduction budget (see page 1 line 54-60). That is another reason why this study should be published. After this eight-years project started in 2006, a series of huge budget flood reduction plans kept conducted but the budget allocation is still mysterious. In order to avoid misunderstanding that this result had been proven as a long-term phenomenon, the topic of this paper can be changed to 'Are the Rich less Prone to Flooding during Typhoon Morakot and Typhoon Fanapi in the Southern Taiwan?'

Other issues had been proposed by reviewers including what type of construction in the project, the definition of flooding, the luxury dwellings during inundations in Taiwan, introducing hedonic price method on flood reduction benefit assessment, the motivation of rich people to reduce flood loss will be added, further explain or revise in the main content.

Sincerely yours,
The corresponding author

| Anonymous Referee #2 | Authors |
| --- | --- |
| Like the previous reviewer, I don't understand the connection between the results obtained and the conclusions made by the authors. How can they be certain that lower flooding probability for high-income groups can be attributed to budget priorities for a flood risk reduction project that was launched in 2006? Like the previous reviewer mentioned, this conclusion could only be supported with additional analyses for floods that occurred before 2006. | It is intuitive that the motivation is the flood risk reduction in their residing areas when the local governments decided the priority and the allocation of public flood protections. However, the advantage of high income people and their political power is difficult to prove because that works under the table. We can only prove that through the outcome. We used the lowest administrative entity (villages) during extreme typhoon cases to have the data |

| | |
|---|---|
| Furthermore, the relationship between income and political power/motivation/advantage has not been proven in this context. | on residents' income and large sample size. Since we need widespread flooding to do this empirical study, the non-extreme typhoon cases are not suitable. Extreme cases seldom happen. Currently, we did not have the flooding probability of villages before the project. However, this study did proof that those 2006 high income (10%) villages had less flooding probability than 2006 non-high income villages during 2009 and 2010 typhoons in Southern Taiwan. Therefore, the topic of this paper can be changed to 'Are the Rich less Prone to Flooding during Typhoon Morakot and Typhoon Fanapi in the Southern Taiwan?'. I may point out this research limitation at the end of this paper. |
| It is not clear at all from the text what type of construction work the flood risk reduction project entailed, and therefore how it may have differed in effectiveness between different income groups. | The budget was mainly for structural flood protection, such as levees, pumping stations, and detention ponds. Almost all rivers already had some sort of levees before the project. Due to the Project, the local governments decided the priority and the allocation of enhancing levees and building detention ponds. The decision process had been described in the manuscript. The content of the Project can be added to the manuscript. |
| The study is missing an investigation of the correlation between house price and income. I note that the propensity scoring matching exercise quantified house prices per ping; this approach will mask overall differences in house prices due to different house sizes. If significant correlation between the | In Taiwan, the flooding is mainly inundation which is caused by extreme rainfall and insufficient drainage rather than river flooding. Even during extreme typhoons like Morakot and Fanapi, most of the casualty was not from flooding (mainly because of landslides). In Taiwan, seismic safety is |

| | |
|---|---|
| variables is found (which I suspect will be the case), this poses a significant issue: a. The authors mention in line 125 of page 4 that "the higher the average house price of a village, the less likely that it will be flooded". So, perhaps higher income areas are less prone to flooding simply because of features directly related to their higher house prices (e.g., better quality construction) rather than any additional flood risk reduction measures implemented in 2006? | emphasized in the commercials of high price buildings rather than flood prevention because the drainage is managed and regulated by the government.

We put the house price in the model and the hypothesis of that is negative because the house price is usually adopted to measure the benefit of public flood protection measures called the hedonic price method. It is a mechanism of cost-benefit analysis which leads public flood protection to the areas where high price buildings are located. Since the risk reduction efforts toward more population and high real estate price areas are democratic and economic (cost-benefit analysis) mechanisms, respectively, rent-seeking is the most possible mechanism. |
| The assumptions of the methodology are not well explained. Flooding is represented as a binary variable, such that very different levels of inundation would be treated identically. This feature is not necessarily a problem, but the authors should address the simplified nature of this assumption and the fact that areas with higher probabilities of flooding are not necessarily those that will experience the most amount of flood damage. Furthermore, no definition of flooding is provided in the text – what is the minimum level of water depth treated as a flood, how is flood depth/extent measured in each village, is there any subjectivity in its | The data sources of flooding investigations of those two typhoons were stated in the manuscript. The process of flooding investigation is that the flooding locations (point) were reported by residents and then the investigation team of each city/county went to check and plotted the flooding area. However, since each team had a different format of records, the flood depth was not recorded in some cities/counties (only areas). The minimum recorded flood depth is 20cm from the team that recorded flood depth. The recorded flood depth will be added to the manuscript. In line 107 of page 4, all villages in Pingtung county, |

| | |
|---|---|
| measurement? How many high- and low-income villages are captured in the analyses? What were the criteria for inclusion of a certain village in the analyses? The answers to these questions should be provided in the text, to understand the reliability of the underlying analyses. | Kaohsiung city, and Tainan city were adopted in this study. There is no criteria for the inclusion of villages. The altitude (elevation) and slop were adopted to control the nature of villages. |

---

## Author Comment (AC5)

Dear 2 anonymous referees and the editor,

I would like to thank your valuable comments. I had posted my responses right after your comments. I want to further explain how can we revise to responses your major concern.

The most concerning issue of suggesting rent-seeking mechanism is that we don't have the flooding probability before 2006. The reason is that we have to get a large sample size to do this empirical study but the wide spared flooding events seldom happen and the affected regions might not the same. However, the 0702 flooding which was caused by 2004 Typhoon Mindulle, and the flooding during 2008 Typhoon Kalmaegi both mainly happened in Southern Taiwan. We may calculate the flooding probability of rich and non-rich communities to show that the flooding probability before 2009. Whether we have the flooding probability before 2006 may not be an issue as well. The Project was the first project funded by the central government to reduce the flood risk in rivers managed by local governments and before the Project started in 2006 all local governments in Southern Taiwan did not have enough flood reduction budget (see page 1 line 54-60). That is another reason why this study should be published. After this eight-years project started in 2006, a series of huge budget flood reduction plans kept conducted but the budget allocation is still mysterious. In order to avoid misunderstanding that this result had been proven as a long-term phenomenon, the topic of this paper can be changed to 'Are the Rich less Prone to Flooding during Typhoon Morakot and Typhoon Fanapi in the Southern Taiwan?'

Other issues had been proposed by reviewers including what type of construction in the project, the definition of flooding, the luxury dwellings during inundations in Taiwan, introducing hedonic price method on flood reduction benefit assessment, the motivation of rich people to reduce flood loss will be added, further explain or revise in the main content.

Sincerely yours,
The corresponding author